# Leveraging Deep Learning for Fine-Grained Categorization of Parkinson’s Disease Progression Levels through Analysis of Vocal Acoustic Patterns

**DOI:** 10.3390/bioengineering11030295

**Published:** 2024-03-21

**Authors:** Hadi Sedigh Malekroodi, Nuwan Madusanka, Byeong-il Lee, Myunggi Yi

**Affiliations:** 1Industry 4.0 Convergence Bionics Engineering, Pukyong National University, Busan 48513, Republic of Korea; hadi_sedigh@pukyong.ac.kr; 2Digital of Healthcare Research Center, Institute of Information Technology and Convergence, Pukyong National University, Busan 48513, Republic of Korea; nuwanmadusanka@hotmail.com; 3Division of Smart Healthcare, Pukyong National University, Busan 48513, Republic of Korea

**Keywords:** Parkinson’s disease (PD), deep learning, transfer learning, speech analysis, mel spectrogram

## Abstract

Speech impairments often emerge as one of the primary indicators of Parkinson’s disease (PD), albeit not readily apparent in its early stages. While previous studies focused predominantly on binary PD detection, this research explored the use of deep learning models to automatically classify sustained vowel recordings into healthy controls, mild PD, or severe PD based on motor symptom severity scores. Popular convolutional neural network (CNN) architectures, VGG and ResNet, as well as vision transformers, Swin, were fine-tuned on log mel spectrogram image representations of the segmented voice data. Furthermore, the research investigated the effects of audio segment lengths and specific vowel sounds on the performance of these models. The findings indicated that implementing longer segments yielded better performance. The models showed strong capability in distinguishing PD from healthy subjects, achieving over 95% precision. However, reliably discriminating between mild and severe PD cases remained challenging. The VGG16 achieved the best overall classification performance with 91.8% accuracy and the largest area under the ROC curve. Furthermore, focusing analysis on the vowel /u/ could further improve accuracy to 96%. Applying visualization techniques like Grad-CAM also highlighted how CNN models focused on localized spectrogram regions while transformers attended to more widespread patterns. Overall, this work showed the potential of deep learning for non-invasive screening and monitoring of PD progression from voice recordings, but larger multi-class labeled datasets are needed to further improve severity classification.

## 1. Introduction

Parkinson’s disease (PD) is a progressive neurodegenerative disorder characterized by motor symptoms like tremors, rigidity, and slowed movement [1,2,3]. However, pathology underlying PD begin years before the clinical diagnosis, with early manifestations like hyposmia, speech disorders, depression, constipation, and sleep disturbances frequently overlooked [4,5]. Diagnosing PD during the initial phase and initiating treatment can potentially impede the rate of progression of this degenerative disorder [6].

While neurological examination methods like the Movement Disorder Society Unified Parkinson’s Disease Rating Scale (MDS-UPDRS) and brain scans are among the main criteria for diagnosing PD, they have limitations such as cost, accessibility, clinician bias, and difficulty monitoring progression and treatment effectiveness [1,2,3,7,8]. Therefore, there is a need for alternative diagnostic approaches that are more objective, cost-effective, and accessible.

Speech difficulties are often one of the initial and most serious signs of PD, severely affecting how patients communicate and their overall quality of life [9]. Over 80% of PD patients have some vocal dysfunction, including decreased volume, lack of tone, reduced fundamental frequency range, slurred speech, or abnormal rhythms and melodies [10,11]. This can occur up to 5 years before motor symptoms like tremors appear [12,13]. While assessing writing and walking needs specialized devices, voice can be captured and analyzed without special equipment or clinic visits [13]. Therefore, speech analysis provides a promising opportunity for early PD detection and continuous monitoring.

Various acoustic analysis techniques including measuring fundamental frequency variation, noise parameters, and non-linear dynamics, have been explored for detecting and quantifying vocal symptoms [14,15]. However, recent research has increasingly focused on leveraging advanced machine learning and neural network approaches to automatically detect PD through speech analysis [16]. Significant work has centered on selecting optimal features for shallow classifiers as well as determining ideal architectures for deep learning classifiers.

The first approach involves hand-crafting acoustic features, including certain variants of the jitter, shimmer, and harmonic-to-noise ratio that are indicative of PD speech impairments [17,18,19,20,21,22] and using traditional machine learning (ML) methods, such as support vector machines (SVM), random forests (RF), k-nearest neighbors (KNN), and regression trees (RT) [20,21,22,23,24,25,26,27].

Mamun et al. tested ten algorithms on 195 vocal recordings, finding that LightGBM, a gradient-boosting method, achieved 95% accuracy in classifying PD [23]. Govindu et al. recently studied early PD detection via telemedicine using ML models on audio data from 30 PD and 30 control subjects. Their RF classifier had the best performance—91.83% accuracy and 0.95 sensitivity for detecting PD [20]. Wang et al. implemented 12 machine learning models on the 401 voice biomarkers dataset to classify subjects as PD or not. They also built a custom deep learning model with a classification accuracy of 96.45% [24]. Pramanik et al. achieved high accuracy in PD detection using Naïve Bayes algorithms [28]. Other studies focused on feature selection techniques. Lamba et al. tested combinations of three selection methods (mutual information gain, extra tree, genetic algorithm) and three classifiers (Naive Bayes, KNN, RF), finding that the genetic algorithm plus RF performed best with 95.58% accuracy [25].

In contrast to the previous approach, which primarily used manual feature engineering and shallow classifiers, the second approach harnesses deep learning to automatically learn features directly from speech data. Various neural network architectures have been designed and tested, including Convolutional Neural Networks (CNNs), Recurrent Neural Networks (RNN) like Long Short-Term Memory Networks (LSTMs) networks, a combination of them, and more recently, transformer-based models. These models directly learn feature representations from the speech signal or spectrograms, including sustained vowels, continuous speech, and repeating syllables. Deep learning models can alleviate the need for expert-crafted features and have achieved state-of-the-art (SOTA) results on PD detection from speech [8].

Aversano et al. developed LSTM and CNN models to analyze voice recordings segmented into 1 s intervals consisting of vowels, phrases, and sentences. These voice samples were transformed into mel spectrogram representations as input to the models, which achieved an F1 score of 97%. However, a notable limitation of this study was that the researchers did not ensure that the training and validation sets were speaker-independent, which could potentially introduce biases and may limit the generalizability of the models’ performance [29]. Similarly, Shah et al. employed a CNN-based model that analyzed 1 s speech chunks transformed into log-scaled mel spectrograms (LMS) for detecting PD from vowel phonations of /a/ and /i/, achieving 90.32% accuracy [30]. Another study employed a MobileNet CNN model with various types of spectrograms as input. The findings indicated that speech energy spectrograms and mel spectrograms yielded the highest accuracy rates of 96% and 92%, respectively [31]. A study by Khojasteh et al. evaluated the performance of a CNN model on sustained vowel phonation recordings of the /a/ lasting over 5 s. When tested on 2 s voice samples segmented into 815 ms frames, the CNNs achieved a classification accuracy of 75.7%. An interesting aspect of their approach involved data augmentation techniques like flipping (vertically and horizontally) and rotating the frames, which were applied to the training dataset. However, since the inputs were spectrogram-based images representing time-frequency information, such spatial transformations may not have been suitable augmentation techniques [8]. Quan et al. employed an end-to-end model incorporating both 2D and 1D CNNs to achieve 92% accuracy in classifying PD based on speech tasks involving the reading of both simple and complex sentences. Their model operated on a sequence of overlapping segments derived from the LMS representation of the input audio. However, the study did not specify the length of this sequence of overlapping segments [10].

Furthermore, some researchers further improve performance by using transfer learning to adapt these speech models, leveraging knowledge already gained on other tasks. Hireš et al. proposed an ensemble approach involving multiple fine-tuned versions of the Xception deep learning model. When applied to a subset of the sustained vowel recordings dataset (PC-GITA), focusing on the vowels /a/, /i/, /o/, /u/, and /e/, this ensemble method achieved an impressive 99% accuracy in classifying the presence of PD based solely on the voice recordings. In their approach, the 1 s voice signal was transformed into a spectrogram, which was then blurred before being processed by the models [13]. In another study, Wodzinski et al. fine-tuned a ResNet architecture model using a subset of the PC-GITA dataset containing only the vowel sound /a/. By transforming the audio recordings into spectrograms, their model achieved an accuracy of over 90% in classifying the presence of PD [11]. More recently, Klempíř et al. found that self-supervised speech models, such as wav2vec which have been pre-trained on 960 h of 16 kHz English speech, generate valuable embeddings for PD detection. These models achieved AUROC (area under the receiver operating characteristic curve) scores ranging from 0.77 to 0.98 across various datasets, which included repeated /pa/ syllables. Notably, this pipeline can be immediately applied to raw audio signal recordings without the need for segmenting [32]. In summary, the deep learning approach shows promise for PD detection from voice, with recent work achieving accuracies over 90% using techniques like CNNs, LSTM models, and self-supervised learning.

Prior studies have focused on binary classification of PD detection from voice recordings, distinguishing between people with PD and healthy controls. However, clinical applications would benefit from more granular subtype classification beyond this binary distinction [33]. In this work, we first explored the use of multi-class classification to detect PD and differentiate between various stages based on their MDS-UPDRS III scores. Part III of the MDS-UPDRS assesses motor function in Parkinson’s disease patients. We trained models to classify voice recordings into three classes. This paper also compared three DL architectures widely used in computer vision tasks. The models were trained using LMS representations derived from sustained vowel phonations from a publicly accessible dataset. Secondly, the study examined how the length of audio clips and particular vowel sounds impacted the effectiveness of these models. Additionally, previous studies segmented audio recordings before analysis but did not evaluate model performance on full recordings; in this work, we applied an ensemble method across segments to obtain overall classifications for entire segments after splitting. Finally, we employed visualization techniques such as Grad-CAM [34] and t-SNE [35] to provide possible explanations of the deep learning model’s predictions, highlighting discriminative regions in the LMS inputs that influence particular classification decisions.

## 2. Materials and Methods

Figure 1 shows the architecture of our speech classification system that categorizes speech signals into one of three classes: healthy, Parkinson’s disease mild, or severe. The system captures the audio signal, preprocesses it into segments, and converts the segments into LMSs—visual representations of audio frequency content over time. These spectrograms are input to a deep neural network that extracts informative audio features. A classifier model then categorizes the speech into one of three classes by matching the extracted features to learned patterns. In essence, the system transforms audio into images, extracts features using deep learning, and classifies speech based on those features.

### 2.1. Dataset

The present study used the Italian Parkinson’s voice and speech database. The dataset comprises speech recordings in the .wav format obtained from Italian individuals diagnosed with Parkinson’s disease, as well as healthy control subjects. This database was collected through the efforts of Dimauro et al., as referenced in [36,37]. Building on prior work that found sustained vowels to be more predictive of Parkinson’s diagnosis compared to words or sentences [19], this study focused its analysis specifically on short vowels. By concentrating only on short vowel samples, potential factors like language and education that could potentially skew the results can be eliminated.

As outlined in Table 1, the subset includes sustained vowel recordings (vowels /a/, /e/, /i/, /o/, and /u/) from 22 healthy controls (12 female, 10 male) and 28 PD patients (9 female, 19 male). The participants were closely matched by age, with an average of 67.1 years (±5.2 years) in the control group and 67.2 years (±8.7 years) in the PD group. The PD patients were further classified by their score on Part III of the MDS-UPDRS. Figure 2 shows the histogram of audio lengths across three groups: Healthy Controls (HC), Mild Parkinson’s Disease (PD_Mild), and Severe Parkinson’s Disease (PD_Severe). Notably, HC samples predominantly fall within approximately 5 s, while PD groups exhibit a broader range.

### 2.2. Data Preprocessing

We performed data preprocessing to convert and structure the raw audio data into an applicable format that could be effectively analyzed via deep learning models. Initially, all audio recordings from the database were resampled at 16 kHz to ensure a consistent sampling rate. Subsequently, recordings with excessive background noise were removed from the dataset during this preprocessing stage (2 healthy participants were excluded for this reason). The total number of audio recordings after this part was 475. The audio clips were also trimmed to remove any leading or trailing silence. The raw speech data contained audio recordings of different lengths, as shown in Figure 2. To create manageable training batches with consistent sample sizes, the recordings were segmented into fixed-length clips (1 s and 5 s), with each segment overlapping the previous one by 50%, padding shorter utterances and truncating longer utterances. The original dataset was processed to create two distinct versions for training purposes. In the First Segment (FS) version, only the first segment from each audio recording was utilized. Alternatively, the All Segments (AS) version encompassed all segments derived from the recordings rather than just the initial segment. These two approaches to segmentation produced different training datasets, FS and AS, from the same raw data. These varying combinations of segmentation approaches and duration made four unique training datasets (FS-1, FS-5, AS-1, and AS-5) from the same raw data (Figure 3). From now on in this paper, these abbreviations will be utilized to reference the particular dataset versions. The details of the modified datasets are provided in Appendix A.

Since the models that were used in this study were suitable for images, after segmenting the voice recordings, they needed to be transformed into an image data format. All recordings were then converted from waveform audio to LMS-based images. The LMS is a representation of an audio signal that accounts for the human auditory perception of frequency and loudness. It is obtained by first computing a spectrogram using the Short-Time Fourier Transform (STFT), which provides the frequency content and amplitude over time, with frequency on a linear Hz scale. The linear frequency axis is then converted to the mel frequency scale using Equation (1):(1)$$m=2595{log}_{10}\left(1+\frac{f}{700}\right)$$
where *m* and *f* represent mel frequency and frequency in mels and Hz, respectively, this conversion results in a mel spectrogram, where the frequency axis is represented in the mel scale, which better approximates the human auditory system’s response to sound frequencies. Finally, the logarithm of the amplitude values (in dB) is taken to mimic the human ear’s logarithmic perception of loudness. The resulting LMS displays the frequency content in mels on one axis and time on the other, with the amplitude represented by a logarithmically scaled color map [38]. In this research, LMS representations were computed using 128 ms (2048 samples) window lengths and 32 ms (512 samples) hop lengths for the STFT, with examples provided in the referenced Figure 4.

Additionally, to reduce overfitting given the initially small training dataset, the limited data set was expanded by applying different types of audio augmentation before executing the voice-to-image transformation process. This data expansion aims to improve generalizability. For this purpose, we performed data augmentation using the torch audio spectrogram augmentation library [39]. Here, various techniques, including time masking, frequency masking, and a combination of them, were applied to each audio and then transformed to the LMS image (Figure 5). Data augmentation was not used for the validation sets, so these sets would resemble real-world data. Finally, the LMSs were resized to 224 × 224 pixels and converted to 3-channel grayscale images for input into the deep learning models.

### 2.3. Training and Deep Learning Models

In this study, we utilized several popular deep learning models for computer vision tasks. Specifically, two popular CNN architectures were employed: ResNet and VGG [40,41]. These CNNs have achieved good performance on benchmark datasets and have become standard models for computer vision. VGG16 and VGG19 are deep convolutional neural network architectures that have 16 and 19 layers, respectively. Both architectures consist of 5 sets of convolutional layers, where each layer is followed by a max pooling layer. The main difference between VGG16 and VGG19 is the number of cascaded convolutional layers in each set. The architecture of VGG16 is shown in Figure 6a. ResNet-50, on the other hand, is a residual network architecture that contains 50 layers (49 convolutional layers organized into 16 residual blocks and one final fully connected layer for output). It utilizes skip connections, which allow the network to skip certain convolutional layers during backpropagation, alleviating the vanishing gradient problem. ResNet-18 is a simplified variant of the original ResNet architecture for image classification. As shown in Figure 6b, it contains 18 layers in total—17 convolutional layers organized into eight residual blocks and one final fully connected layer for output [40,41,42].

In recent years, transformers have become the predominant model architecture for natural language processing (NLP) tasks due to their continuously improving efficiency [43]. The capabilities of transformers are not limited to NLP, though they have also shown excellent skill in image recognition. Architectures like the Vision Transformer (ViT) [44] demonstrate how transformers can match or even surpass CNNs on computer vision datasets. Building on the concepts of ViT, the Swin Transformer [45] introduces a hierarchical design for greater efficiency and the flexibility to model at a variety of scales [43]. We also employed the Swin Transformer architecture in this study to take advantage of its state-of-the-art capabilities. The Swin Transformer model is a pure transformer architecture model that is becoming a general-purpose backbone for various tasks. There are four Swin Transformer configurations: Swin_t, Swin_s, Swin_b, and Swin_l [45]. The Swin_s and Swin_b were chosen as feature extractors in this study. The numbers of parameters for them are 49.6 M and 87.8 M, respectively, as shown in Table 2. The overall architecture of the Swin Transformer is illustrated in Figure 6c. The Swin_s and Swin_b models differ primarily in the size of the embeddings and the number of heads used in their transformer architectures. Swin_b has larger embeddings and more heads than Swin_s. Further details about these models can be found in the original paper [45].

These models have already been trained on a large-scale labeled dataset. The performance metrics of these models on the ImageNet dataset are presented in Table 2. During the training phase, the pre-trained weights (the weights obtained when a model was trained on the ImageNet dataset) were utilized. Transfer learning was applied by tuning the pre-trained layers. The weights learned on ImageNet provide a much better initialization for many computer vision tasks than random weights [46].

The classification layers of the original models were removed and replaced with new classification head. This new classifier uses a neural network with two dense layers before the final classification layer. The first dense layer has 256 neurons, and the second dense layer has 128 neurons (Figure 6d). After each dense layer, a dropout with a probability of 0.5 was applied. This same classification architecture was utilized across all models in the study.

### 2.4. Experimental Setups and Evaluation Criteria

Our implementation leveraged various Python libraries such as PyTorch [39] for deep learning model development, Pandas [47] and NumPy [48] for data analysis, and Matplotlib [49] and Scikit-learn [50] for visualization and some analysis tasks.

As detailed in Table 3, key training hyperparameters used during model optimization included learning rate, batch size, and number of epochs. The models were trained using an Adaptive Moment Estimation optimizer with Weight Decay (AdamW), an optimization algorithm with cross-entropy loss to measure prediction error. A learning rate of 0.0003 was set initially and adjusted over time per a scheduler. We implemented the experiments using a system comprising an Intel Core i7-11700K CPU @ 3.60 GHz, with 128G of RAM and GPU NVIDIA RTX 3090 24G.

This study followed two approaches for classifying audio samples and training the models. The first approach involved segmenting the audio clips into 1 and 5 s segments, as AS-1 and AS-5 methods explained in Section 2.1, thereby increasing the dataset size. The second approach only used the first segments of each audio clip, FS-1 and FS-5. We evaluated whether the segmentation helped improve model accuracy compared to using only the first segmented part.

This study utilized four main evaluation criteria: Precision, Recall, F1 score, and Overall accuracy. Precision refers to the percentage of positive classifications that were correct. Recall (also called sensitivity) measures the percentage of actual positives that were correctly identified. The F1 score combines precision and sensitivity by taking their harmonic mean. Finally, overall accuracy is simply the percentage of total classifications that were correct out of all classifications made.

To calculate these performance metrics, we determined the numbers of true positives (TP), false positives (FP), and false negatives (FN) per class. A TP represents a correct prediction for a given class. An FP is an incorrect prediction that wrongly predicted that class. An FN is a case that belongs to that class but was incorrectly excluded.
(2)$$Accuracy=\frac{TP+TN}{TP+TN+FN+FP}$$
(3)$$\begin{array}{c} Precison=\frac{TP}{TP+FP} \end{array}$$
(4)$$\begin{array}{c} Recall=\frac{TP}{TP+FN} \end{array}$$
(5)$$\begin{array}{c} F1=\frac{2\;\times \;Recall\;\times \;Precison}{Precison\;+\;Recall} \end{array}$$

## 3. Results and Discussion

In this section, we will describe and discuss our results in detail while evaluating the studied models’ performance.

### 3.1. Classification Performance

A stratified patient-independent three-fold cross-validation approach was utilized for all experiments, where the data was partitioned into three folds with no patient overlap across folds to avoid data leakage and reduce potential biases in model evaluation. The model was trained on two folds and evaluated on the held-out fold, and this was repeated three times so that each fold served as the evaluation set once. This ensured a rigorous assessment of model performance on unseen data. We decided not to use a separate test set due to the small database size. To mitigate potential issues caused by an imbalance of class distribution, we utilized the train-time oversampling technique to achieve a more balanced class distribution [51,52].

The cross-validated performance metrics, including precision, recall, F1 score, and accuracy, for each model are presented in Table 4 and Table 5. Additionally, Figure 7 depicts a graphical representation of the cross-validated classification accuracy for each model. In addition, performance by two additional recent architectures were compared in Appendix A.

Table 4 highlights that utilizing the first 5 s of each recording results in higher classification accuracy across all models compared to using only the initial 1 s segment. While all models demonstrate strong performance in correctly identifying HC subjects, they face challenges distinguishing between varying degrees of PD severity. The FS-5 dataset exhibited superior performance in classifying the different stages of PD. When considering only the recognition of HC subjects, the Swin_s model slightly outperformed other models, demonstrated the best performance in terms of precision (97.00 ± 4.24), recall (100.00 ± 0), and F1 score (98.33 ± 2.36). However, its performance showed minimal deviation compared to the other models.

Our findings indicated that the models demonstrated better accuracy when using longer phonation samples as input. As shown in Table 5, models trained on complete audio segments, rather than just the initial segment, exhibit higher average accuracy on 5 s datasets (AS-5). However, this improvement comes at the cost of increased performance variability, as evidenced by larger standard deviations. Notably, the Swin Transformer models demonstrate the largest gain of around 3% when utilizing the AS-5 dataset. In contrast, for the 1 s dataset, particularly the ResNet models, there is no improvement when using the AS dataset. Among the tested models, VGG19 experiences the most significant boost on the 1 s dataset when trained on all segments compared to just the initial segments.

Overall, utilizing complete audio clips for training tends to improve model accuracy, especially for longer 5 s datasets, although this benefit is less pronounced on the shorter 1 s dataset (AS-1). In addition, visual inspection of bar plots in Figure 7 suggests that, for the specific task we have, the deeper architectures do not demonstrate a substantial improvement in accuracy when compared to their shallower counterparts. Furthermore, the transformer-based model showed noticeable performance gains when trained on the AS dataset. Conversely, the CNN-based models evaluated did not exhibit significant improvements from utilizing the full segmented data.

The proposed models for 5 s datasets were evaluated using cumulative confusion matrices and receiver operating characteristic (ROC) curves across three-fold cross-validation. The confusion matrices aggregated results across folds to showcase the overall model performance. Color bars accompanying the confusion matrices illustrated the proportions of observations within each class that were correctly or incorrectly classified, with values ranging from 0 to 1. The ROC curves plotted the trade-off between the true positive rate and the false positive rate, depicting the diagnostic capability of the models. A One versus Rest (OvR) method constructed the ROC curves. The area under the ROC curve (AUC) signified model performance, with higher values indicating better classification ability. Across models, the AUC for the HC class approached 1.00 (Figure 8 and Figure 9), demonstrating strong identification of healthy subjects. For PD classes, VGG16 achieved slightly higher AUCs compared to other models. Furthermore, the analysis revealed an increase in the AUC from the FS to the AS dataset, particularly for the PD_Mild class, with a 4% improvement. This suggests that the models exhibited slightly better discrimination capabilities when utilizing the full-segment dataset. Furthermore, the transformer-based models exhibited higher AUC values when trained on the larger AS-5 dataset, suggesting that these models benefited from the increased data availability for improved classification performance.

The analysis of the confusion matrices in Figure 8 and Figure 9 suggests that the models excel at accurately identifying samples from the HC group, exhibiting the highest precision and recall for this class. For the FS-5 dataset, there were no instances where an HC sample was incorrectly predicted as PD_Severe or vice versa. However, some instances labeled PD_Severe were misclassified as PD_Mild, and vice versa, indicating potential challenges in distinguishing between these two classes. To better evaluate the VGG16 model’s accuracy for different vowels, we grouped the results by the sustained vowel present in the dataset. The confusion matrices for each vowel are shown in Figure 10. Of the vowels, /u/ had the highest recall for HC and PD_Severe groups (100%) while having a lower recall value for the PD_Mild group (75%).

Although binary classification was not employed in this study, we combined the results to compare accuracy with previous works that utilized the Italian-speaking Parkinson’s speech dataset. Specifically, we categorized HC as negative and all PD cases as positive. The accuracy results of this binary classification are summarized in Table 6.

These results are promising; however, recent studies [53,54] indicated that the models employed for pathological voice detection are typically trained using small-scale data, hindering their ability to perform consistently across diverse datasets. As a result, the performance of these models fluctuates considerably depending on the dataset encountered. This is largely due to the scarcity and variability in the quality of medical voice recordings available for training such systems [54]. This can limit model robustness compared to speech recognition systems trained on ample large-scale datasets. For greater generalizability and diagnostic precision, more consistent and substantial medical voice datasets are required.

**Table 6 bioengineering-11-00295-t006:** Comparison of accuracy results obtained on the Parkinson Italian speaking dataset.

Author	Model	Accuracy [%]
Aversano et al. [29]	LSTM	97.1
Klempíř et al. [32]	Wav2Vec	95.0
Hireš et al. [54]	Xception	97.8
Toye et al. [17]	SVM	98.9 ^1^
Current study	Swin_s	98.5 ± 2.50
Current study	VGG16	98.1 ± 3.23

^1^ Using hand-crafted features.

In previous studies [11,29] on PD classification using audio recordings, researchers have typically segmented the recordings into smaller parts before extracting features and training machine learning models. The researchers assessed the models’ performance on the segmented audio excerpts and reported the corresponding results for these segments. However, they did not provide performance results for complete audio samples. This study employed a simple ensemble method to enable a fair evaluation and comparison of different audio segmentation approaches. Specifically, we passed each segment through the trained model to get a prediction, then took the most common predicted class across all segments as the final prediction for the recording, effectively using majority voting. This allows the comparison of different segmenting approaches equally in terms of overall recording classification. After using this approach, we calculated the cumulative confusion matrix and accuracy, as shown in Figure 11 for the AS-5 dataset. This is a more realistic test scenario, as in real-world applications, we would need to make predictions on individual audio. When implementing this approach, the accuracy of the VGG19 model increased by around 1% compared to results on the AS-5 dataset. Accuracy for the other models did not change significantly or even decreased slightly for this dataset. Despite overall lower performance compared to not using ensembling, our dataset still achieved slightly higher accuracy than when we used the FS dataset, especially when leveraging transformer-based models. This increases more pronounce for the AS-1 dataset that is shown in Appendix A.

We further evaluated the segmentation and ensemble approach by applying it separately to each individual vowel sound, aiming to determine which vowel benefited the most from this technique and achieved the highest performance. The results summarized in Table 7 and Table 8 demonstrate that the vowels /u/ and /o/ may have the greatest ability among the models to differentiate between Parkinson’s classes. Notably, the findings suggest that when utilizing solely the vowel /u/ for classification with the VGG16 model, an impressive F1 score of 96% can be attained. The performance on vowel /u/ in [29] contributed to the overall improved accuracy across the different methods utilized. These results align with earlier findings in [55] that the vowel /u/ had the highest classification accuracy out of the vowels /a/, /o/ and /u/ tested. Rusz et al. [15] provided further support, identifying abnormalities in vowel articulation and acoustics, such as reduced vowel space area, among PD patients, especially for the vowel /u/.

### 3.2. Grad Cam Feature Visualization

Grad-CAM (Gradient-weighted Class Activation Mapping) is a visual explanation technique for CNNs [34]. Grad-CAM utilizes the gradient information from the final convolutional layer of a CNN to generate a heat map representing the regions of the input image that are most relevant for the network’s prediction. Specifically, it computes the gradients of the target concept (i.e., the class output) with respect to the feature maps of the last convolutional layer. By pooling these gradients over the spatial dimensions, Grad-CAM produces a coarse localization map that highlights the parts of the image that have the greatest influence on CNN’s decision [34,38]. The architecture explaining the Grad-CAM technique is shown in Appendix A.

The Grad-CAM feature map visualizations in Figure 12 represent three 5 s audio clips of the vowel sound /o/ from the FS-5 dataset. To maintain consistency, we exclusively used data from the second fold of the FS-5 dataset and the corresponding trained models.

The generated heatmaps highlighted the specific regions in an LMS input image that significantly influenced the model’s prediction. A comparison of the visualization results across different columns revealed key differences between the CNN-based and Swin transformer-based architectures. The CNN models demonstrated more localized attention, focusing on specific local areas in the images [56]. In contrast, the visualizations for the Swin transformer network displayed attention that was more scattered and less spatially localized.

The models generally placed less emphasis on the higher frequency components of the LMSs, particularly in the range greater than 1024 Hz, suggesting that these regions were less discriminative for the classification task. However, it was noteworthy that the Swin Transformer models, in addition to their focus on lower frequencies, less than 512 Hz, also exhibited sensitivity to relatively higher frequencies when detecting healthy control subjects. Furthermore, the ResNet 18 model for the healthy control class demonstrated primary activation in the high-frequency range.

When examining the temporal patterns for the healthy class, it was evident that CNN models primarily focused on the first half to the middle of the audio clips, while transformer-based models were more consistent across time frames. For the mild class, models generally concentrated on the middle period. For the severe class, VGG16 displayed a distinct pattern compared to the other studied models. This model was activated on the middle frequency range (around 2048 Hz) and the timeframes of the initial segments. Additionally, there was a moderately intense region towards the end of the spectrogram. In contrast, the other models focused more on the second half of the audio clips and lower frequencies.

Additional visualizations showcasing Grad-CAM feature maps are presented in Appendix A.

This suggests that the network heavily relies on the spectral patterns in this specific time-frequency region, indicating that the network is also considering some higher-frequency components.

### 3.3. Analyzing Feature Extraction Capability

In the previous section, Grad-CAM visualizations demonstrated qualitative differences between the features extracted by different architectures on our classification FS-5 dataset. To further analyze these representations, the t-distributed Stochastic Neighbor Embedding (t-SNE) technique can be utilized to project high-dimensional feature spaces into a 2D representation, allowing for visualization and interpretation of the learned representations.

Figure 13 presents 2D scatter plots that visualize the distribution of features extracted from the layer just before the classifier in each model. Each class is represented by a different color, allowing for visual analysis of how well the features separate the classes prior to classification.

The t-SNE visualization clearly shows three distinct clusters corresponding to the Healthy, PD_Mild, and PD_Severe classes across all models. Architectures like VGG16, Swin_s, and ResNet50 exhibit cleaner separations between these class clusters, suggesting their ability to extract more discriminative features from the log mel spectrogram images. Notably, the ResNet50 model forms the most compact clusters, indicating higher feature similarity within each class. However, there is some overlap between the PD_Mild and PD_Severe classes, particularly in the region where their feature points intersect. This overlap suggests that certain mild and severe cases may share similar feature characteristics, making it challenging to distinguish them based solely on the extracted features.

Despite the subtle overlap between PD_Mild and PD_Severe classes, all models successfully separated the Healthy class from the Parkinson’s disease classes, demonstrating the effectiveness of using log mel spectrogram images for distinguishing between healthy and Parkinson’s voices.

## 4. Conclusions

This study explored multi-class classification of Parkinson’s disease from speech recordings using deep learning approaches. Several popular CNN and transformer models were trained on log mel spectrogram representations of sustained vowel recordings to categorize samples as healthy controls, mild, or severe Parkinson’s disease labeled based on their MDS-UPDRS III scores. The models demonstrated strong capabilities to distinguish healthy samples from those with Parkinson’s, achieving over 95% precision. However, they struggled to reliably differentiate between mild and severe Parkinson’s, with classification precision closer to 85%. The findings revealed that models performed better when utilizing longer speech segments. The Swin transformer architecture attained the best accuracy in terms of binary classification, though its superiority over CNNs was marginal for this task. Considering overall accuracy, VGG16 can be proposed as the best model with 91.8%. Applying ensemble techniques across segments and focusing analysis on vowels, /u/ and /o/ recordings further improved accuracy by 1–4%. Moreover, visualization methods highlighted discriminative regions and features learned by models, showing transformers identify more widespread patterns while CNNs focus on localized spectrogram areas.

A key limitation of this study was the relatively small dataset size, which may have impacted the models’ ability to reliably distinguish between mild and severe cases of Parkinson’s disease. The limited availability of large-scale, well-annotated medical datasets can hinder the generalization capabilities of such models for real-world clinical applications.

In conclusion, this work demonstrates the potential of leveraging deep learning techniques on spectrogram inputs derived from voice recordings to enable non-invasive detection and monitoring of different stages of Parkinson’s disease progression. However, to further enhance the identification of disease severity from patient voices, our future work will focus on building larger multi-class labeled datasets of Parkinson’s cases. Additionally, further research could explore a broader range of SOTA architectures and input representations beyond log mel spectrograms, potentially enhancing the classification accuracy.

## Figures and Tables

**Figure 1 bioengineering-11-00295-f001:**
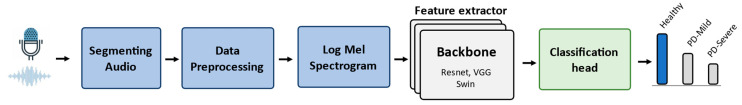
The workflow diagram of our classification system.

**Figure 2 bioengineering-11-00295-f002:**
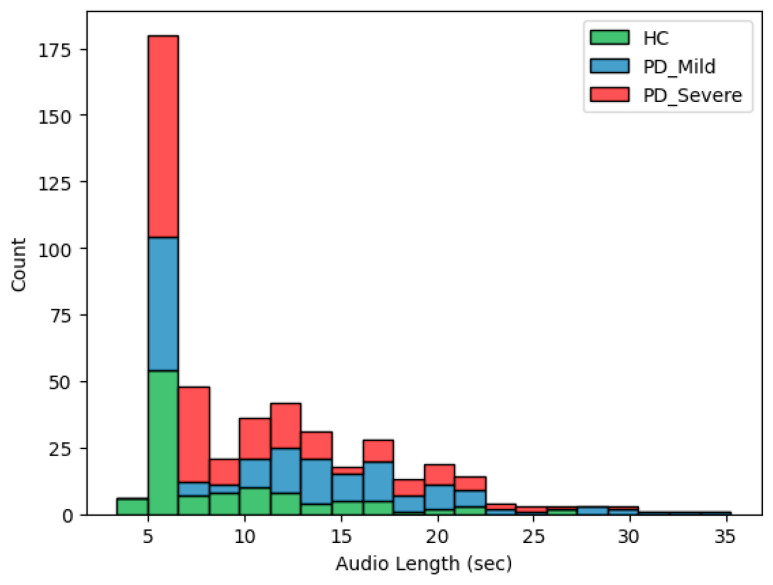
The histogram illustrates the distribution of audio lengths across three groups: HC, PD_Mild, and PD_Severe. Most audio samples are around 5 s in length, with a count exceeding 150.

**Figure 3 bioengineering-11-00295-f003:**
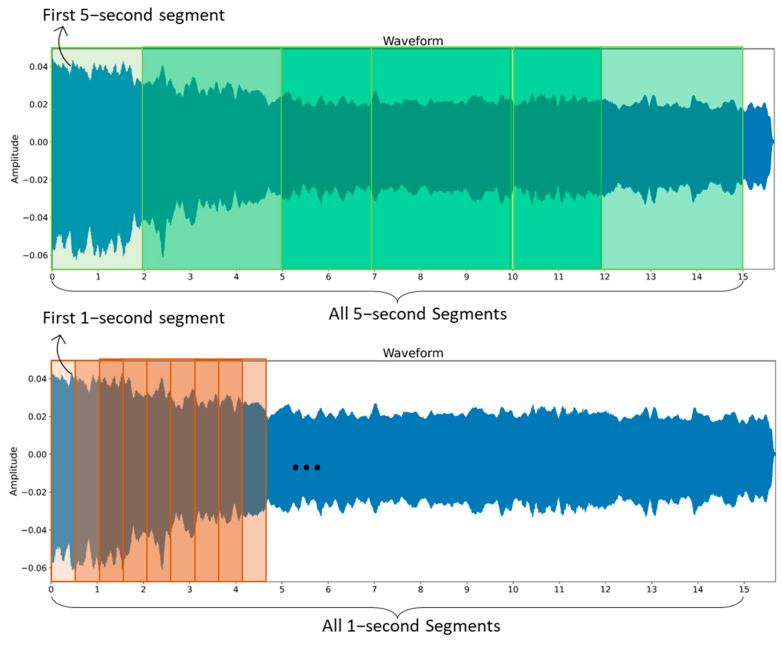
Overview of the process used to construct distinct datasets from the original dataset.

**Figure 4 bioengineering-11-00295-f004:**
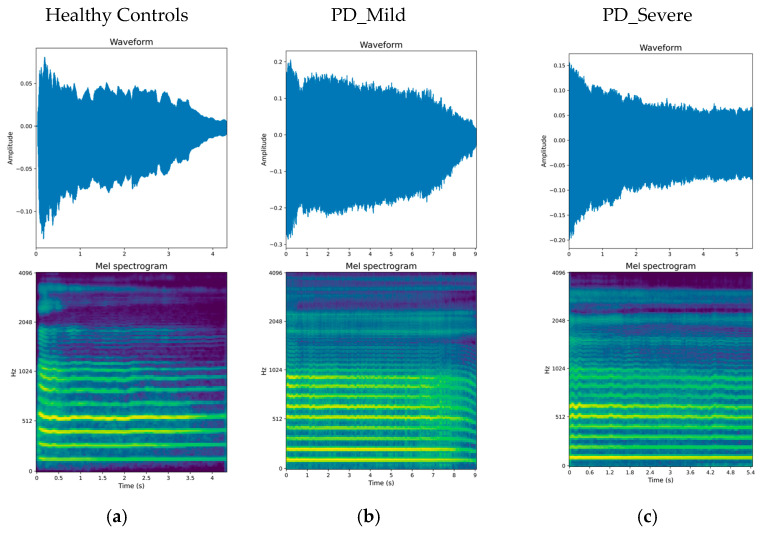
Speech sound examples. The upper panel in each example shows the acoustic waveform. The lower panel shows the corresponding log mel spectrogram representation (128 mel-bands).

**Figure 5 bioengineering-11-00295-f005:**
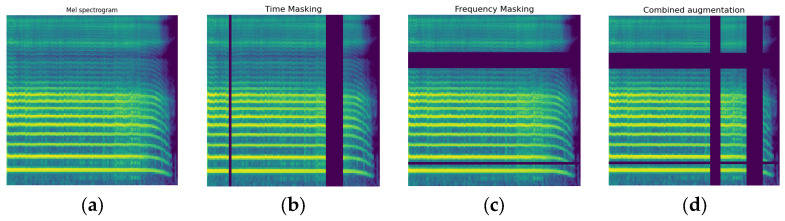
The effects of data augmentations on LMSs: (**a**) displays the original LMS without any augmentations; (**b**) shows the LMS with time masking applied, which masks blocks of time steps. This forces the model to rely more on context; image (**c**) shows the LMS with frequency masking applied, which masks blocks of frequencies; and (**d**) demonstrates the combination of these augmentations.

**Figure 6 bioengineering-11-00295-f006:**
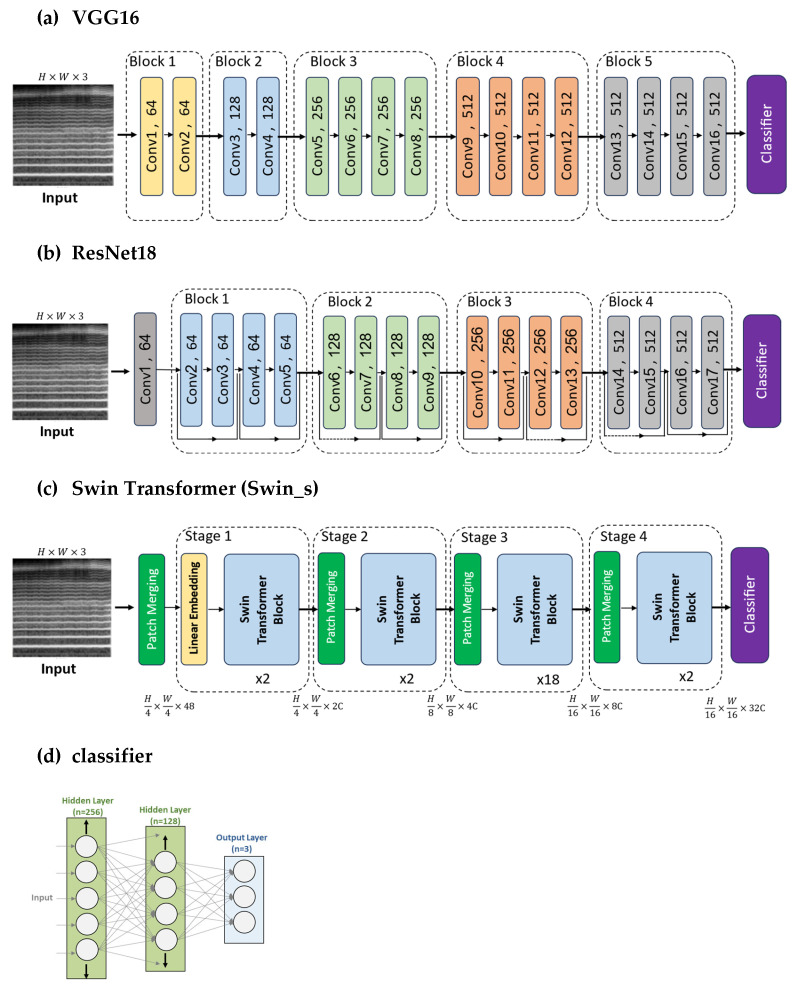
Overview of the architecture of models used in this research.

**Figure 7 bioengineering-11-00295-f007:**
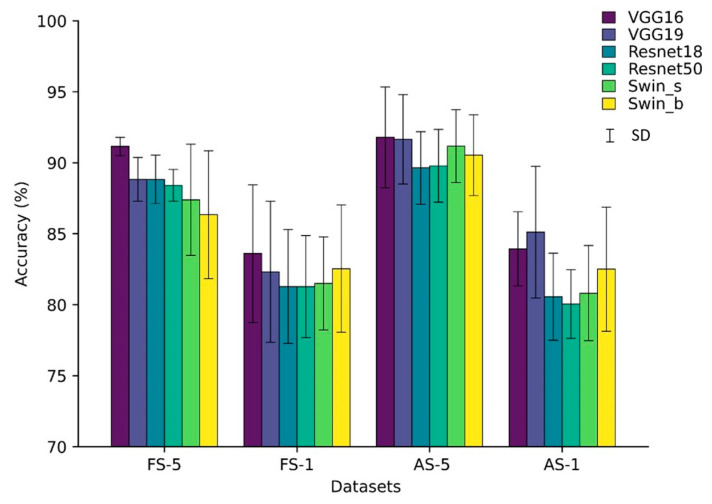
Bar chart showcasing the average accuracy of studied models across modified datasets, with error bars representing the standard deviation (SD). For a clear comparison, the accuracy scale begins at 70%.

**Figure 8 bioengineering-11-00295-f008:**
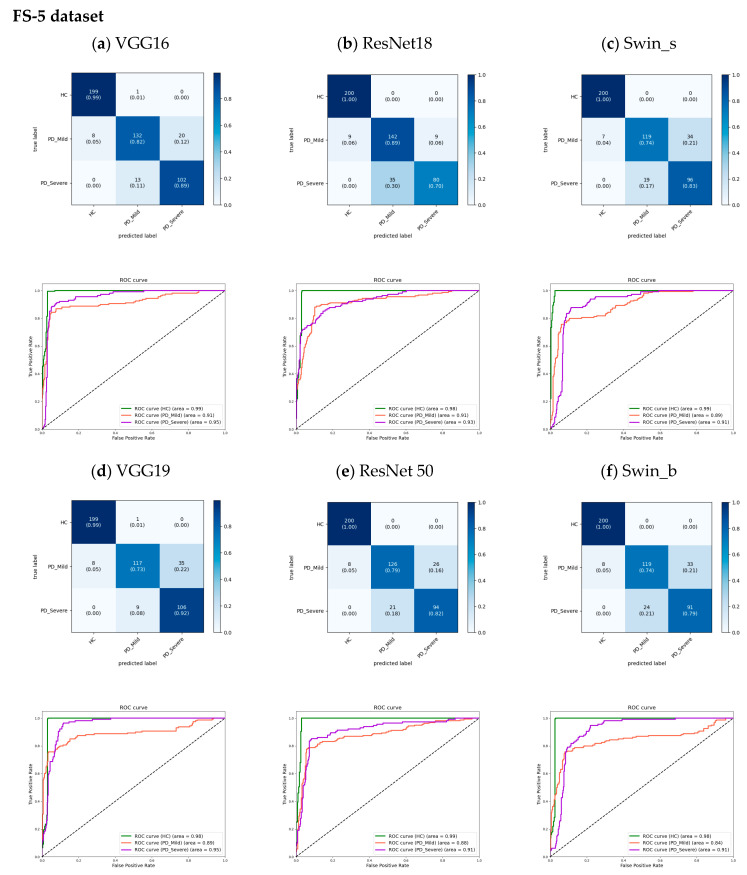
The cumulative confusion matrices and ROC curves show the performance of each model across three folds of cross-validation on the dataset limited to only the FS-5 dataset.

**Figure 9 bioengineering-11-00295-f009:**
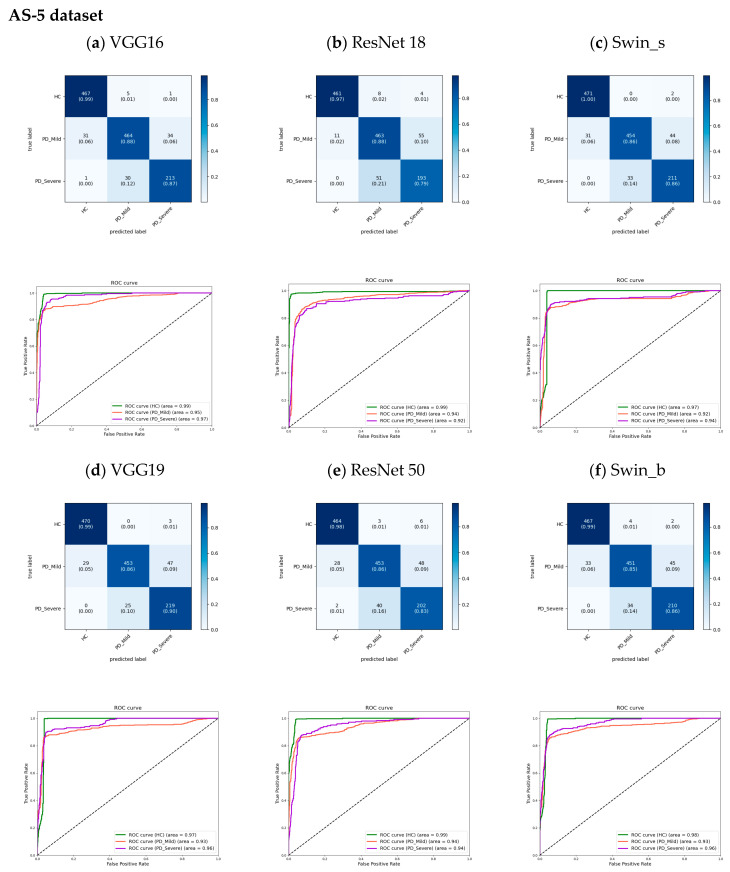
The cumulative confusion matrices and ROC curves show the performance of each model across three folds of cross-validation on the dataset limited to the AS-5 dataset.

**Figure 10 bioengineering-11-00295-f010:**
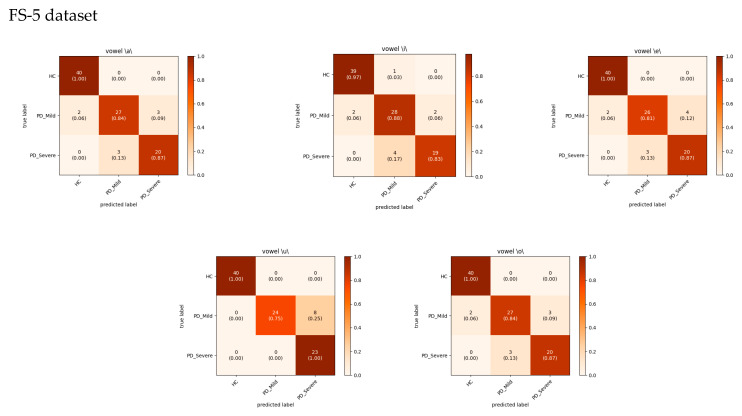
The cumulative confusion matrix for each sustained vowel recording for the VGG16 model. Color bars display the proportion of observations within each class that were correctly or incorrectly classified, with values ranging from 0 to 1.

**Figure 11 bioengineering-11-00295-f011:**
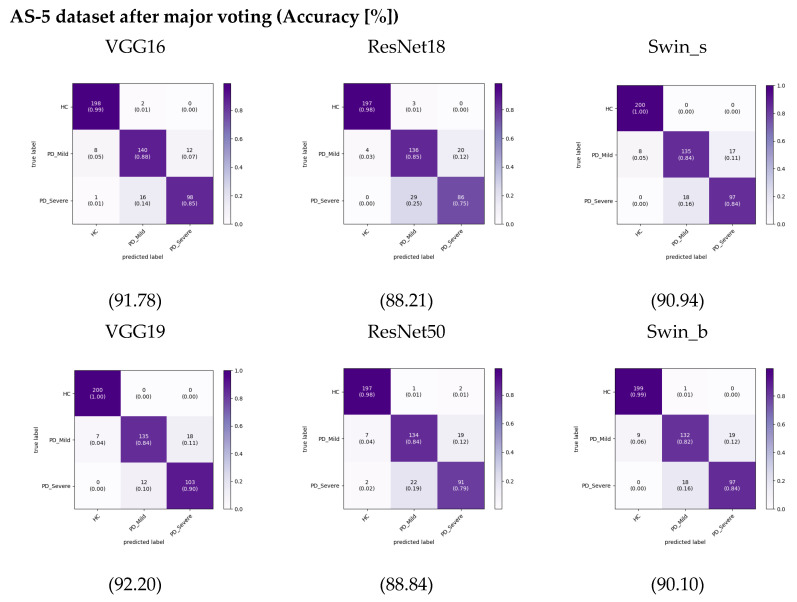
Cumulative Confusion matrix for each model after applying majority voting to predictions on the AS-5 dataset. Color bars display the proportion of observations within each class that were correctly or incorrectly classified, with values ranging from 0 to 1.

**Figure 12 bioengineering-11-00295-f012:**
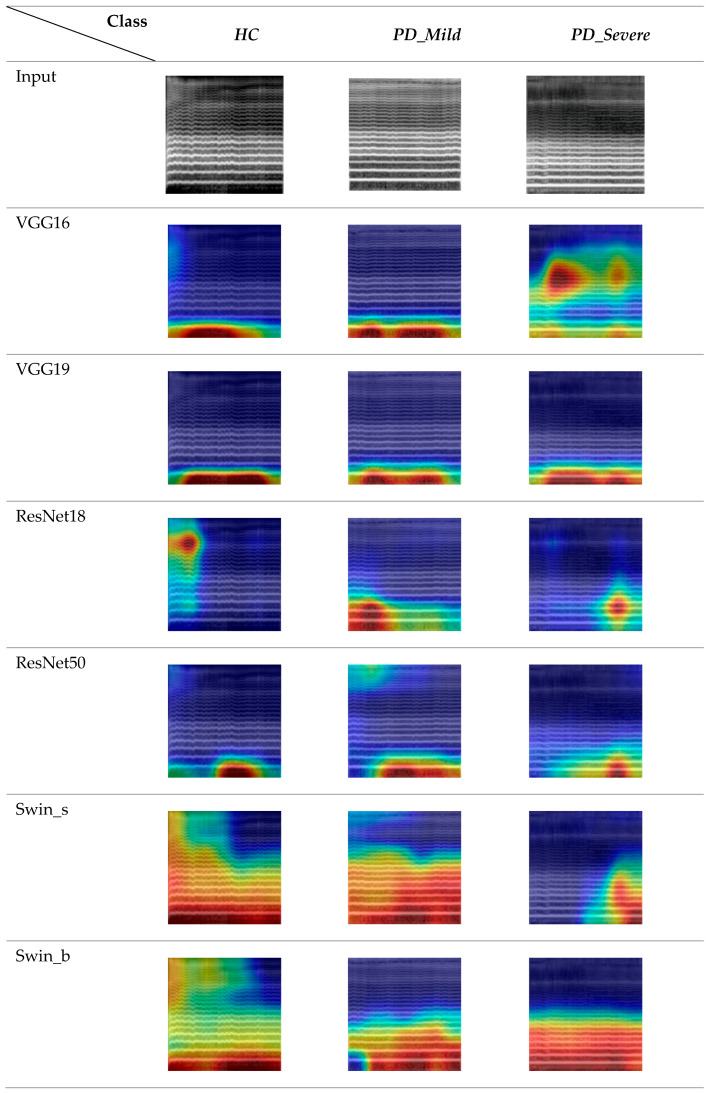
Grad-CAM visualization features different models across various classes for specific vowel /o/.

**Figure 13 bioengineering-11-00295-f013:**
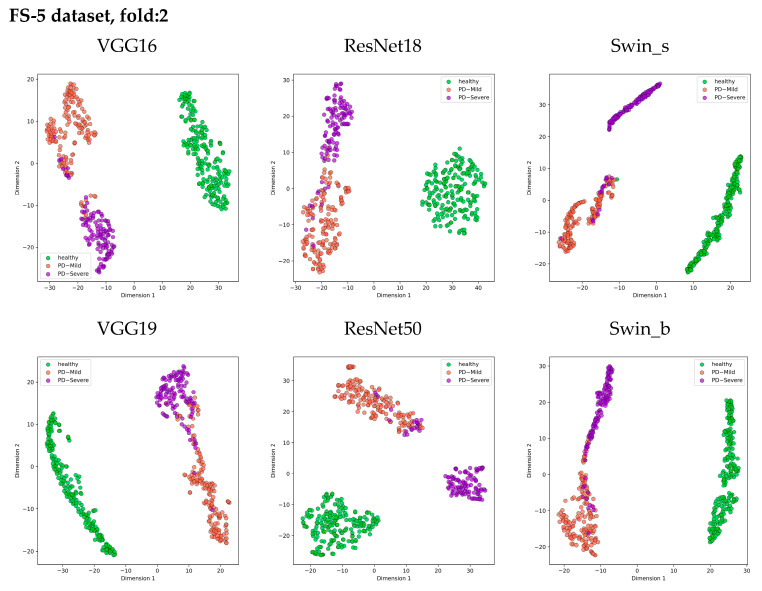
Visualization of feature space in 2D using t-SNE for each model.

**Table 1 bioengineering-11-00295-t001:** Demographic information, including gender, and age ranges of the dataset.

Class	MDS-UPDRS III	Subjects		Age	
		Male	Female	Male	Female
Healthy	~	10	12	60–72	60–77
PD_Mild	1–10	7	3	50–77	40–63
PD_Severe	11–24	12	6	65–75	54–80

**Table 2 bioengineering-11-00295-t002:** Presents the architectural details of the ResNet, VGG, and Swin Transformer models employed in this study, along with their respective performances on the ImageNet-1K dataset. All these models were designed to process input images with dimensions of 224 × 224 pixels.

Model	acc@1	acc@5	#params
ResNet18	69.758	89.078	11.7 M
ResNet50	76.13	92.862	25.6 M
VGG16	71.592	90.382	138.4 M
VGG19	72.376	90.876	143.7 M
Swin_s	83.196	96.36	49.6 M
Swin_b	83.582	96.64	87.8 M

**Table 3 bioengineering-11-00295-t003:** Parameter settings for training models.

Parameter	Values
Image size	224 × 224 pixels
# Epochs	100
# Batch-size	64
Initial Learning Rate	3 × 10^-4^
Optimizer	AdamW (β_1_ = 0.9, β_2_ = 0.999, Weight decay = 0.01)
Loss	Cross entropy

**Table 4 bioengineering-11-00295-t004:** Cross-validated classification performance (mean ± SD) for each model using the FS datasets. The table compares precision, recall, F1-score, and accuracy across models.

FS Datasets			Models					
		Metric (%)	VGG16	VGG19	ResNet18	ResNet50	Swin_s	Swin_b
5 s	HC	Precision	96.67 ± 4.71	96.67 ± 4.71	96 ± 4.32	96.67 ± 4.71	**97.00 ± 4.24**	96.67 ± 4.71
		Recall	99.67 ± 0.47	99.33 ± 0.94	**100.00 ± 0**	**100.00 ± 0**	**100.00 ± 0**	**100.00 ± 0**
		F1 score	98.00 ± 2.83	98.00 ± 2.16	98.00 ± 2.16	**98.33 ± 2.36**	**98.33 ± 2.36**	98.33 ± 2.36
	PD_Mild	Precision	91.00 ± 6.38	**92 ± 3.56**	80.33 ± 1.25	86.00 ± 7.07	88.67 ± 7.72	83.33 ± 3.09
		Recall	82.67 ± 8.06	73.00 ± 3.74	**88.67 ± 5.73**	79.33 ± 6.6	74 ± 22.45	73.67 ± 21.17
		F1 score	**86.00 ± 1.63**	81.00 ± 3.56	84 ± 2.94	82.33 ± 2.05	77.67 ± 12.5	76.33 ± 12.97
	PD_Severe	Precision	84.67 ± 8.96	75.33 ± 4.5	**89.67 ± 7.72**	80.00 ± 10.42	77.67 ± 13.27	75.67 ± 9.81
		Recall	**88.67 ± 9.74**	**92.33 ± 3.09**	69.33 ± 2.49	82.33 ± 8.34	83.67 ± 13.6	79.00 ± 8.29
		F1 score	**85.67 ± 5.91**	82.67 ± 1.89	78.33 ± 4.64	80.33 ± 2.62	78.33 ± 0.47	76.33 ± 1.25
		Accuracy	**91.15 ± 0.64**	88.84 ± 1.54	88.83 ± 1.71	88.41 ± 1.13	87.39 ± 3.92	86.34 ± 4.50
1 s	HC	Precision	96.00 ± 4.32	93.67 ± 3.3	95.33 ± 4.5	96.33 ± 4.5	95.00 ± 3.56	**96.67 ± 2.49**
		Recall	99.67 ± 0.47	99.67 ± 0.47	**100.00 ± 0**	99.33 ± 0.47	98.33 ± 2.36	97.67 ± 0.94
		F1 score	97.67 ± 2.62	**96.67 ± 2.05**	97.33 ± 2.36	97.67 ± 2.62	97.00 ± 1.63	97.00 ± 0.82
	PD_Mild	Precision	75.67 ± 10.4	78.33 ± 10.08	74.00 ± 8.52	**80.67 ± 7.13**	74.33 ± 9.46	74.33 ± 9.98
		Recall	**77.33 ± 2.49**	65.67 ± 18.45	67.67 ± 7.76	62 ± 23.15	72.67 ± 8.22	75.00 ± 6.68
		F1 score	**76.33 ± 6.6**	70.33 ± 13.02	70.67 ± 7.93	66.33 ± 15.08	72.67 ± 6.02	74.67 ± 8.26
	PD_Severe	Precision	**73.67 ± 4.03**	69.00 ± 6.48	65.67 ± 0.94	65.00 ± 8.52	69.67 ± 4.5	69.00 ± 7.87
		Recall	64.67 ± 16.11	73.00 ± 16.31	67.33 ± 9.46	**76.33 ± 14.38**	63.67 ± 17.56	67.67 ± 11.15
		F1 score	67.67 ± 9.29	69.67 ± 9.18	66.00 ± 5.35	**69.00 ± 2.16**	64.33 ± 8.96	68.00 ± 7.87
		Accuracy	**83.60 ± 4.85**	82.33 ± 4.97	81.29 ± 4.01	81.28 ± 3.59	81.50 ± 3.27	82.55 ± 4.48

Boldfaced values indicate the best performance for each metric.

**Table 5 bioengineering-11-00295-t005:** Cross-validated classification performance (mean ± SD) for each model using the AS datasets. The table compares precision, recall, F1-score, and accuracy across models.

AS Datasets			Models					
		Metric (%)	VGG16	VGG19	ResNet18	ResNet50	Swin_s	Swin_b
5 s	HC	Precision	94 ± 7.79	94.67 ± 7.54	**97.67 ± 3.3**	94 ± 7.07	94.33 ± 7.32	94 ± 8.49
		Recall	98.67 ± 1.25	99.33 ± 0.94	97.67 ± 1.89	98 ± 1.41	**99.67 ± 0.47**	98.67 ± 0.47
		F1 score	96 ± 3.56	96.67 ± 4.71	**97.33 ± 1.25**	96 ± 4.32	96.67 ± 4.03	96 ± 4.24
	PD_Mild	Precision	92.67 ± 3.30	**95 ± 4.08**	88.67 ± 3.09	91 ± 2.94	93.33 ± 3.77	92 ± 2.16
		Recall	**87.67 ± 5.44**	85.67 ± 4.5	**87.67 ± 3.3**	85.67 ± 4.64	86 ± 2.45	85.33 ± 6.6
		F1 score	**90.33 ± 4.19**	90 ± 3.27	88 ± 2.94	88.33 ± 4.03	89.33 ± 3.3	88.67 ± 3.68
	PD_Severe	Precision	**87 ± 8.83**	82.67 ± 7.41	76.67 ± 2.36	79 ± 8.16	83.67 ± 10.21	82.67 ± 7.13
		Recall	87.67 ± 6.18	**90 ± 8.16**	79.33 ± 5.91	82.67 ± 6.6	86.67 ± 8.26	86.33 ± 5.44
		F1 score	**86.33 ± 3.68**	85.33 ± 1.25	78 ± 3.74	80.67 ± 6.18	84.33 ± 0.94	84 ± 1.63
		Accuracy	**91.80 ± 3.55**	91.66 ± 3.15	89.64 ± 2.55	89.79 ± 2.57	91.18 ±2.56	90.54 ± 2.85
1 s	HC	Precision	95.33 ± 6.6	93.67 ± 8.96	93.67 ± 3.3	90.67 ± 6.24	**95.67 ± 2.87**	93.33 ± 7.32
		Recall	95.33 ± 3.09	**99.33 ± 0.47**	98.33 ± 1.25	96.67 ± 2.87	97.33 ± 1.7	97 ± 0.82
		F1 score	95 ± 3.27	96 ± 4.24	95.67 ± 2.05	93.33 ± 4.03	**96.33 ± 2.49**	95 ± 3.56
	PD_Mild	Precision	82 ± 2.16	**85.33 ± 7.41**	78.33 ± 4.11	78.33 ± 2.36	80.33 ± 5.19	83.67 ± 4.92
		Recall	79.33 ± 6.94	**79.67 ± 12.66**	77 ± 11.34	72.33 ± 5.31	72.33 ± 9.43	74 ± 16.67
		F1 score	80.33 ± 3.4	**81.67 ± 6.55**	77.33 ± 4.92	75.33 ± 3.30	75.67 ± 5.31	76.67 ± 8.73
	PD_Severe	Precision	**69 ± 6.38**	71 ± 8.60	58.33 ± 7.59	63.67 ± 13.02	57.33 ± 8.5	66.33 ± 10.4
		Recall	71 ± 4.08	68.67 ± 17.52	54.33 ± 18.55	63.67 ± 10.37	66.33 ± 18.37	**71.67 ± 13.72**
		F1 score	**69.67 ± 3.68**	69 ± 12.83	54.67 ± 11.79	63.67 ± 11.26	61 ± 11.58	67.33 ± 6.80
		Accuracy	83.94 ± 2.61	**85.12 ± 4.64**	80.57 ± 3.07	80.05 ± 2.42	80.82 ± 3.35	82.51 ± 4.38

Boldfaced values indicate the best performance for each metric.

**Table 7 bioengineering-11-00295-t007:** The average F1 score for each model grouped by sustained vowels only for the first segment datasets.

FS Datasets		Models (Avg F1 Score [%])					
	Vowel	VGG16	VGG19	ResNet18	ResNet50	Swin_s	Swin_b
5 s	/a/	91	95	90	89	86	85
	/i/	90	87	88	91	86	85
	/e/	90	88	85	85	86	85
	/o/	91	90	88	87	85	86
	/u/	92	83	92	88	93	90
1 s	/a/	80	77	75	80	75	80
	/i/	85	84	85	82	83	83
	/e/	82	82	82	83	82	85
	/o/	82	83	81	79	81	84
	/u/	86	84	82	82	85	80

**Table 8 bioengineering-11-00295-t008:** The average F1 score for each model grouped by sustained vowels for all segment datasets after applying major voting.

AS after Major Voting		Models (Avg F1 Score [%])					
	Vowel	VGG16	VGG19	ResNet18	ResNet50	Swin_s	Swin_b
5 s	/a/	90	90	85	88	88	89
	/i/	90	92	89	92	90	90
	/e/	91	90	86	88	87	88
	/o/	90	93	88	87	95	92
	/u/	96	96	92	89	94	91
1 s	/a/	86	77	75	80	83	84
	/i/	85	84	85	82	79	82
	/e/	82	82	82	83	80	83
	/o/	90	83	81	79	84	86
	/u/	83	84	82	82	82	88

## Data Availability

The datasets used in this study are publicly available. For the Italian Parkinson’s voice and speech database, please visit https://ieee-dataport.org/open-access/italian-parkinsons-voice-and-speech (accessed on 10 March 2024). The dataset was provided by Giovanni Dimauro from Università degli Studi di Bari. The source code is also available on request.

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
