# Peer review of "Leveraging Deep Learning for Fine-Grained Categorization of Parkinson’s Disease Progression Levels through Analysis of Vocal Acoustic Patterns"

_bioengineering, 2024, doi:10.3390/bioengineering11030295_

Round 1

Reviewer 1 Report

Comments and Suggestions for Authors

Leveraging Deep Learning for Fine-Grained Categorization of Parkinson’s Disease Progression Levels through Analysis of Vocal Acoustic Patterns

I found the introduction well written and informative, with, however, more pros than cons. Despite this, it follows a standard introduction schema that guides also inexpert reader to the hypotheses and methods.

In the first paragraph of the introduction, the authors stated: “ Pathology emerges years before 38 diagnosis, with early symptoms frequently overlooked” This sentence is not clear. However, I advise to give a more clear definition of this statement about the first symptoms.

In the introduction, I appreciated the fact that the authors made brief reviews of previous findings about the alterations in the voice of PD and the data analysis methods used to discover these alterations (CNN, etc.)

However, I advise you to briefly add (one sentence) about other early alterations in PD. Similarly, (in one sentence) the literature about voice alteration and methods to analyze it. According to me, this could represent a brief compendium to your introduction.

“Our results demonstrate that deep learning models can make acceptable distinctions be-116 yond binary classification in this domain when provided with multi-class labeled data” In the introduction, this type of statements need to be avoided. These sentences are good in a discussion, but not in the intro.

In the methods, more info about the dataset is needed.The dataset contains speech recordings from Italian Parkinson’s disease patients and healthy 135 controls, originally described in” This sentence needs to be completed. I like the fact that the dataset participant demographics were added in the main text and table.

The figure 1 is informative and clear as well as fig.2

Line 196- “In this research, LMS representations were computed using 128 ms window lengths and 32 ms hop lengths for the STFT, with examples provided in the referenced Figure 4.” The 128 msec of window length should be motivated. Why this specific length?

However, the procedure is well explained and the authors have given a step-by-step description that can allow replicability of the methods. This is also extremely detailed, despite the length of the manuscript.

In Figure 7 the error bars are omitted in the figure legend. Please add.

“The main takeaway”- please correct this expression. Similarly, “the analysis of bar plots” there is not a specific analysis to be done on a bar plot. I mean that “visual inspection “ Should sound better.

Figure 10-11 legend- please add info about colored bars.

I suggest renaming the section “results” as “Results and Discussion”.

I advise to add specific limitations in the conclusions. 

Reviewer 2 Report

Comments and Suggestions for Authors

Authors need to address the following suggestions. 

1. The novelty of the proposed work need to be explained clearly in the abstract. Application of pretrained SOTA models cannot be considered as novelty.

2. Why analysis is tried with three networks namely ResNet, Vgg and swin transformer alone ?  From the recent literature, it could be inferred that, networks like DenseNet and EfficientNet yield optimal results. Authors need to compare the results with all SOTA networks.

3. Intrepretation for GradCAM and t-SNE plots need more clarity. 

4. Introduction section need to be reorganized, having a clear hierarchy and logical connect in the methods discussed.

Comments on the Quality of English Language

Minor changes

Round 2

Reviewer 1 Report

Comments and Suggestions for Authors

Please check the manuscript for typos. The authors addressed my concerns. 

Author Response

We have checked the typos again. Thank you.

Reviewer 2 Report

Comments and Suggestions for Authors

Authors have not addressed this comment. 

Why analysis is tried with three networks namely ResNet, Vgg and swin transformer alone?  From the recent literature, it could be inferred that, networks like DenseNet and EfficientNet yield optimal results. Authors need to compare the results with all SOTA networks.

Comments on the Quality of English Language

Moderate changes

Author Response

Thank you for the comment. We evaluated three widely used network architectures, ResNet, VGG, and recently Swin Transformer, to establish a strong baseline for our analysis. Our goal was not an exhaustive evaluation of every available network architecture. However, we acknowledge the merit in your suggestion, and a comparative study involving a broader set of SOTA networks could be a valuable extension of this work.

However, following the reviewer's valuable feedback, we extended our analysis to include two additional subtypes of SOTA network architectures, DenseNet and EfficientNet. We trained and evaluated DenseNet121 and EfficientNetB0 on our first 5-second (FS-5) dataset using the same experimental setup as the previously reported. The results are summarized below and added to the Supporting Information.

FS datasets

Models

Metric (%)

VGG16

VGG19

Dense121

Eff_b0

Swin_s

5 sec

HC

Precision

96.67±4.71

96.67±4.71

96.67±4.71

96.67±4.71

97.00±4.24

Recall

99.67±0.47

99.33±0.94

99±1.41

100±0

100.00±0

F1 score

98.00±2.83

98.00±2.16

97.67±2.05

98.33±2.36

98.33±2.36

PD_Mild

Precision

91.00±6.38

92±3.56

76±2.94

79.67±9.03

88.67±7.72

Recall

82.67±8.06

73.00±3.74

79.33±16.44

79.33±6.6

74±22.45

F1 score

86.00±1.63

81.00±3.56

76.67±9.29

79.67±7.76

77.67±12.5

PD_Severe

Precision

84.67±8.96

75.33±4.5

78±12.73

76.33±8.38

77.67±13.27

Recall

88.67±9.74

92.33±3.09

67.67±4.19

71.33±12.76

83.67±13.6

F1 score

85.67±5.91

82.67±1.89

72±2.94

73.33±8.96

78.33±0.47

Accuracy

91.15±0.64

88.84±1.54

84.8±44.09

86.09±8.22

87.39±3.92

As the results demonstrate, while DenseNet121 and especially EfficientNetB0 achieved competitive performance, they did not outperform the top-performing model from our initial analysis and showed high variability.

In future iterations, we plan to explore additional architectures on our planned new dataset, including different subtypes of DenseNet and EfficientNet, to assess their potential benefits for our task. We appreciate the constructive feedback, as it aligns with our commitment to rigorous evaluation and continuous improvement.